# Is live birth rate decline from frozen cycles performed during Chinese new year holiday season: A single center retrospective cohort

**Jiajia Zhai**[1☉], **Songping Yi**[2☉], **Junyan Zhang**[3], **Aimin Li**[1], **Shibin Zhao**[4], **Bozheng Zhang**[5], **Guimin Hao**[1]*

**1** Department of Reproductive Medicine, The Second Hospital of Hebei Medical University, Shijiazhuang, China, **2** Department of Obstetrics, Guangdong Second Provincial General Hospital, Guangzhou, Guangdong, China, **3** Department of Clinical Epidemiology and Evidence-Based Medicine, Shanxi Bethune Hospital, Shanxi Academy of Medical Sciences, Third Hospital of Shanxi Medical University, Tongji Shanxi Hospital, Taiyuan, China, **4** Department of Reproductive Medicine, The Fourth Hospital of Hebei Medical University, Shijiazhuang, China, **5** Department of Pediatrics, College of Arts and Sciences, Emory University, Atlanta, Georgia, United States of America

☉ These authors contribute equally to this work.
* haoguimin@hebmu.edu.cn

## Abstract

### Purpose

This study investigates the impact of the Chinese New Year (CNY) holiday on frozen embryo transfer (FET) outcomes in assisted reproductive technology (ART), particularly in vitro fertilization (IVF). Previous research has highlighted the negative effect of the CNY holiday on fresh embryo transfer outcomes, prompting an exploration of whether FET outcomes are similarly affected.

### Methods

A retrospective cohort study was conducted at the Second Hospital of Hebei Medical University, analyzing FET cycles performed from January 2012 to December 2022. A total of 4456 women were included, with 305 undergoing FET during the CNY holiday season and 4151 during non-holiday periods. The primary outcome measure was the live birth rate. Multivariate logistic regression and propensity score matching (PSM) were applied to assess the differences between the CNY and non-CNY (N-CNY) groups.

### Results

Multivariate logistic analysis revealed no significant difference in live birth outcomes between the CNY and N-CNY groups (OR = 1.11, 95% CI 0.86 to 1.42). PSM analysis further confirmed that live birth rates were similar between the groups, with 39.0% in the N-CNY group and 44.2% in the CNY group (p = 0.208). These findings suggest that FET outcomes are not adversely affected by the CNY holiday.

**Data availability statement:** Data is available from https://www.openicpsr.org/openicpsr/project/231961.

**Funding:** The 2025 Annual Medical Science Research of Health Commission of Hebei Province (20250555). The Appropriate Health Technology Project for County-Level Comprehensive Hospitals in Hebei Province (20210018); the Medical Applicable Technology Tracking Project in Hebei Province (GZ2023030); Hebei Provincial Government Funded Clinical Medicine Excellent Talent Program (2022).

**Competing interests:** The authors have declared that no competing interests exist.

**Abbreviations:** AMH, Anti-Müllerian hormone; ART, assisted reproductive technology; ART, assisted reproductive technology; CI, confident interval; CNY, Chinese New Year; DR, down regulation; E2, estrogen; FET, frozen embryo transfers; FSH, follicle-stimulating hormone; GnRH, gonadotropin-releasing hormone; hCG, human chorionic gonadotropin; HRC, hormone replacement cycles; HRT, hormone replacement treatment; ICSI, intracytoplasmic sperm injection; IVF, in vitro fertilization; LBR, live birth rate; LH, luteinizing hormone; NC, natural cycle; OR, odd ratio; OSC, ovulation stimulation cycle; P, progesterone; PCOS, polycystic ovary syndrome; PPOS, progesterone-primed ovarian stimulation; PRL, prolactin; PSM, propensity score matching; T, testosterone.

## Conclusion

FET outcomes during the CNY holiday season remain resilient, contrasting with findings from fresh embryo transfer studies. This may be attributed to the shorter duration and less intensive preparation of FET cycles, reducing the influence of sociocultural events and psychological stress. Further multicenter studies are needed to validate these findings and explore the impact of other significant sociocultural events on ART outcomes.

## Background

The Chinese New Year is China's largest festival, spanning a month of nationwide celebrations [1]. Our team previously studied the impact of the Chinese New Year holiday on the outcomes of fresh embryo transfers of in vitro fertilization(IVF) [2]. The research indicates that the Chinese New Year holiday may reduce the live birth rate of fresh embryo transfers in IVF. Though the Chinese New Year holiday has a negative impact on fresh embryo transfers, it is not clear whether it affects the pregnancy outcomes of frozen embryo transfers.

Some studies suggest that frozen embryo transfers have a higher live birth rate (LBR) compared to fresh embryo transfer in high responders, but there is no overall difference in cumulative pregnancy rates [3–5]. For freezing embryo transfer, there is no evidence that the time from egg retrieval to embryo transfer affects live birth rates in a freeze-all strategy [6,7]. It is well-known a fresh cycle typically takes 10–60 days from ovulation stimulation to transplantation, while a frozen cycle usually spans 10–30 days [8]. The preparation time for a frozen cycle is notably shorter than that for a fresh cycle [9]. While fresh stimulation cycles are susceptible to social psychological factors, despite the shorter duration, frozen cycles may also be influenced by social and cultural factors [10]. The two main steps of in vitro fertilization are ovarian stimulation, egg retrieval process, and embryo transfer process. No studies have shown which of these two steps is more affected by psychological stress. Comparing the outcomes of frozen embryo transfer during the Spring Festival period and in normal times is aimed at studying the impact of psychological stress on the embryo transfer process. Further research may be needed to gain a deeper understanding.

## Methods

Inclusion Criteria involve patients who had their initial Assisted reproductive technology (ART), including in vitro fertilization and intracytoplasmic sperm injection (ICSI), underwent a first frozen embryo transfer (FET) cycle from January 2012 to December 2022, and experienced a live birth from the cycle.

Exclusion criteria pertain to patients lacking complete data for relevant variables.

### Implementation procedure

All patients undergoing their first ART cycle at our center had their first FET cycle from January 2012 to December 2022. Controlled ovarian hyperstimulation

comprised various protocols, including Gonadotropin-Releasing Hormone (GnRH) antagonist protocol, luteal phase GnRH agonist protocol, follicular phase long-acting GnRH agonist protocol, Progesterone-Primed Ovarian Stimulation (PPOS) and natural cycle protocol. After oocyte retrieval, all the embryos were frozen. One or more menstrual cycle(s) later, the patient visits for frozen embryo transfer.

In the FET cycle, for endometrium preparation, there are four main treatment cycles in our center:

1. Hormone replacement treatment (HRT): Take oral estrogen (Progynova, Bayer, Leverkusen, Germany) 3 mg, two times daily was commenced on the first or second day of menstruation with the aim of supporting endometrial proliferation and suppressing follicle growth. After 7–9 days, a vaginal ultrasound examination was performed to confirm that no dominant follicle had emerged and to measure endometrial thickness. If the endometrial thickness is less than 8 mm, the estrogen dosage was raised to 8 mg daily, and/or vaginal administration of estradiol (Femoston, Abbott B.V.) can be administrated, and ultrasound examination was repeated after 4–8 days. Estrogen for a period of 9–20 days, then add progesterone 60 mg administered as a daily intramuscular injection combined with dydrogesterone (Abbott B.V.) tables 10 mg two times daily for four days, and embryo thaw and transfer was planned.

2. Ovulation Stimulation Cycle (OSC): Initiate treatment on the second day of menstruation by administering 2.5 mg or 5 mg of letrozole daily for five days. Subsequently, assess the stimulation effect via vaginal ultrasound. When the dominant follicle reaches 16–20 mm in diameter, administer 250 mg of Human Chorionic Gonadotropin (hCG) (Ovitrelle, Merck, Kenilworth, USA) to trigger ovulation.

3. Natural cycle (NC): Patients participating in a modified natural cycle for frozen-thawed embryo transfer underwent ultrasound assessments of the dominant follicle from days 10–12 of their menstrual cycle. Ultrasound monitoring continued until the dominant follicle attained a diameter of 16–20 mm. At this point, 250 mg of hCG (Ovitrelle, Merck, Kenilworth, USA) was administered subcutaneously to induce ovulation.

4. Down-Regulation (DR) plus HRT: Administer a long-acting GnRH agonist at a dose of 3.75 mg for pituitary downregulation. Thirty days later, regular HRT was commenced to prepare the endometrium.

After the endometrial lining is prepared, embryo transfer is performed. Both cleavage and blastocyst stage embryos were allowed for transfer in this study. The time of thawing and transferring was based on the developmental stage at the time of freezing. In cleavage stage embryos, thawing was performed on the fourth after progesterone initiation or after hCG injection. Blastocyst embryos were thawed on the sixth day after progesterone initiation or hCG injection. Embryo scoring was performed after thawing according to standard validated morphological characteristics. This standard was based on the ESHRE Istanbul consensus on embryo assessment [11]. The transfer was performed on the day of thawing.

Dates at oocyte retrieval and ET for frozen transfers during the study period were categorized by meteorological season: winter (December, January, and February); spring (March, April, and May); summer (June, July, and August); and fall (September, October, and November).

Then, the luteal-phase support was administered 90 mg progesterone gel (Merck Serono) plus 20 mg dydrogesterone twice daily for the artificial cycle. Dydrogesterone 10 mg twice daily for natural cycle. Based on the embryo transfer dates, participants were divided into the Chinese New Year holiday season (the entire January and February) group (CNY group) and the non-holiday season (months other than Jan and Feb) group (N-CNY group).

## Data retrieval

Data were collected retrospectively from the hospital information system (HIS) database, which included patients' age, classification of infertility, duration of infertility, treatment protocol, and education level. All subjects in the study had undergone testing for their baseline serum levels of follicle-stimulating hormone (FSH) (μIU/mL), luteinizing hormone (LH) (μIU/mL), estrogen (E2) (pg/mL), progesterone(P)(ng/mL), testosterone(T)(ng/mL), prolactin (PRL) (ng/mL), and Anti-Müllerian

hormone (AMH) (ng/mL). The tests were performed using commercial kits (Siemens Healthcare Diagnostics) on an automated chemiluminescence immunoassay analyzer. Treatment protocols for superovulation and numbers of retrieved oocytes were also obtained from the Hospital information system database.

The causes of infertility can be categorized into four main factors:

1. Tubal Factor: This includes tubal obstructions, absence of fallopian tubes, and other issues related to the fallopian tubes that cause infertility.

2. Male Factor: This refers to male infertility caused by severe oligoasthenospermia or azoospermia.

3. Other Factors: These include causes of infertility not covered by the first two factors, such as polycystic ovary syndrome (PCOS), endometriosis, and other female-related factors.

4. Multi Factors: A combination of the aforementioned three factors.

### Outcomes

The main focus was on achieving live birth, characterized by the successful delivery of at least one neonate in a specific embryo transfer cycle. Clinical pregnancy, identified through sonographic evidence of fetal cardiac activity 30 days post-embryo transfer, was considered a secondary outcome, encompassing ectopic pregnancy. Other outcomes taken into account were miscarriage and pregnancy loss before reaching 20 weeks.

### Statistical analysis

Continuous variables exhibiting a normal distribution were represented by the mean and standard deviation, while those with abnormal distribution were characterized by the median and interquartile range (Q1-Q3). Student's t-test and Wilcoxon rank-sum test were applied for normally and abnormally distributed quantitative data, respectively. Categorical variables underwent analysis using Chi-square or Fisher's exact test. Propensity Score Matching (PSM) was conducted to align patients in the N-CNY group with those in the CNY group. The propensity score dataset was constructed through a multi-variable logistic regression model, considering variables such as age, infertility duration, education, FSH, E2, P, PRL, LH, T, AMH, and infertility classification. Caliper matching with a caliper of 0.02 of the pooled standard deviation of the logit of the propensity score was employed. Patients in the CNY group were matched in a 1:1 ratio to patients in the N-CNY group. Multivariate logistic regression, adjusted for variables with a p-value no greater than 0.10, assessed the association between the live birth rate and the two treatment groups for the primary endpoint. Adjustments were made for variables with clinical relevance to the primary endpoint, irrespective of the p-value. Results were presented as an adjusted odds ratio (OR) with 95% confidence interval (CI). Sensitivity analysis was performed on the PSM dataset. All hypothesis tests were two-sided, and statistical significance was considered for p-values<0.05. Sample size calculation adhered to the events per variable principle of multivariable analysis [12], with a minimum of five [13] and a maximum of ten to fifteen endpoint events for each variable [13]. With a live birth occurrence ranging from 40% [14] to 50% [15], and considering the need to adjust ten variables, a target of 150 live births was set. Under these circumstances, 600 samples were deemed necessary.

Stata SE 13 (Serial number 40130), R software version 4.2.0 (http://cran.r-project.org/), and easy-R (www. empower-stats.com) were used for statistical analysis. GraphPad was used to generate figures.

## Results

### Participant enrollment

The study initially screened medical records of 11,000 individuals (Fig 1) undergoing fertility treatments. After applying the exclusion criteria, 4,456 records were deemed eligible for inclusion in the study. These were further classified based on

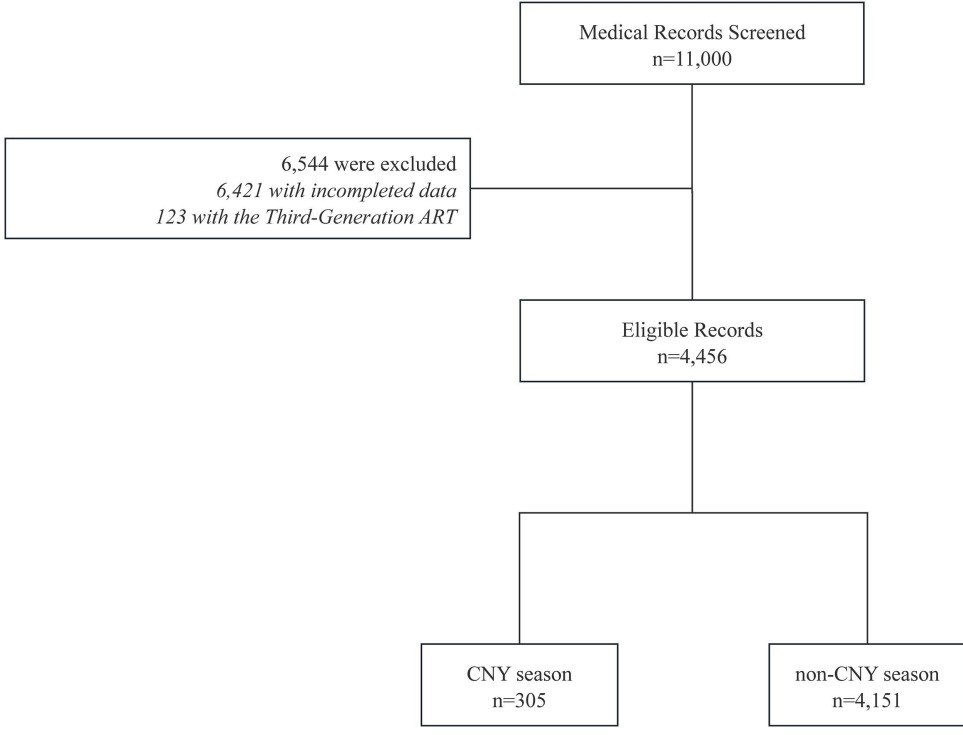

**Fig 1. The flow diagram of participants.**

the timing of the fertility treatment relative to the CNY group. A total of 305 treatments occurred during the CNY season, while the majority, 4,151 treatments, took place outside of the CNY season. The lowest LBR by the month when ET was performed appeared in Mar(40.8%), while the highest LBR was in Feb (51.6%), while the LBR with ET in all other months ranged between 41.0% and 47.4% (Fig 2).

## Demographic and baseline clinical characteristics

The study comprised 4,456 participants, divided into N-CNY (n = 4,151) and CNY (n = 305) groups. Analysis of demographic data revealed no significant differences in the mean age of female participants (N-CNY: 30.87 ± 4.46 years vs. CNY: 30.72 ± 4.73 years, p = 0.579) or male participants (N-CNY: 31.50 ± 4.78 years vs. CNY: 31.60 ± 4.93 years, p = 0.716). The educational attainment across groups showed slight differences, but these were not statistically significant (Table 1).

The median duration of infertility was consistent across both groups at 3.30 years, with interquartile ranges showing slightly wider variation in the CNY group (1.80–5.20 years for N-CNY vs. 2.30–5.20 years for CNY, p = 0.090). In terms of infertility classification, there was no significant difference in the proportion of primary (N-CNY: 53.75% vs. CNY: 57.05%, p = 0.264) and secondary infertility (N-CNY: 46.25% vs. CNY: 42.95%) (Table 1).

Analysis of the etiology of infertility showed no significant differences between the groups. The proportions of cases attributed to tubal factors were similar (N-CNY: 58.23% vs. CNY: 55.74%, p = 0.122), and similarly minor differences were observed in cases attributed to male factors (N-CNY: 0.53% vs. CNY: 0.98%) and multi-factorial causes (N-CNY: 2.10% vs. CNY: 0.66%). However, other factors were slightly more prevalent in the CNY group (N-CNY: 39.15% vs. CNY: 42.62%) (Table 1).

**Fig 2. Live birth rate by the implantation month.**

Hormonal assessments indicated no significant differences in median AMH levels (N-CNY: 3.99 ng/mL vs. CNY: 3.87 ng/mL, p = 0.570), E2, and FSH levels across the groups. However, PRL levels were significantly higher in the CNY group (13.61 ng/mL vs. 12.17 ng/mL, p = 0.001), and T levels also showed a significant difference (N-CNY: 0.51 ng/mL vs. CNY: 0.54 ng/mL, p = 0.026) (Table 1).

## Treatment protocols and outcomes

The majority of participants underwent HRT cycle, with no significant difference between the N-CNY and CNY groups (85.71% vs. 85.57%). OSC was minimally used, observed in only 1.44% of all participants, with slightly less frequent use in the CNY group (0.98%). NC, where no hormonal stimulation is used, and the natural ovulation process is monitored, was also infrequently utilized (N-CNY: 2.72% vs. CNY: 2.62%). DR plus HRC was used in 10.14% of the total cohort, with a slightly higher preference in the CNY group (10.82%). The composition of treatment strategies between these two groups was similar, with a p-value of 0.927 (Table 2).

The GnRH antagonist protocol was adopted by 32.43% of all participants, with the CNY group showing a slightly higher usage rate of 37.70%. The study data showed that 36.47% of participants used the luteal phase GnRH agonist protocol, with a lower adoption in the CNY group (30.16%). The follicular phase long-acting GnRH agonist protocol was used by 25.00% of participants, with similar adoption rates across both study groups. The PPOS was implemented among 5.43% of participants, with very close percentages between these two groups. The compositions of the protocol were comparable between the two groups, with a p-value of 0.103 (Table 2).

Seasonal variation in oocyte retrieval demonstrated significant disparities; particularly, retrievals were more common in the fall and winter for the CNY group (Fall: 55.41%, Winter: 32.13%) compared to the N-CNY group (Fall: 22.43%, Winter: 15.83%, p < 0.001) (Table 2).

The evaluation of high-quality embryos across 4,456 participants revealed no significant difference in embryo quality between the N-CNY and CNY groups (p = 0.087). In the N-CNY group, 48.40% of participants had zero high-quality embryos compared to 46.89% in the CNY group. The proportion of participants with one high-quality embryo was similar across groups (N-CNY: 30.21%, CNY: 29.18%). However, a slightly higher proportion of participants in the CNY group had

**Table 1. Demographics and laboratory test results for all participants.**

| | | All (n = 4456) | N-CNY (n = 4151) | CNY (n = 305) | Statistics | *P*-value |
|---|---|---|---|---|---|---|
| Female age | Mean(SD) | 30.86(4.48) | 30.87(4.46) | 30.72(4.73) | t = 0.56 | 0.579 |
| Female education | < 6 years | 140 (3.14%) | 124 (2.99%) | 16 (5.25%) | χ²=6.24 | 0.101 |
| | 6~9 years | 1316 (29.53%) | 1233 (29.70%) | 83 (27.21%) | | |
| | 10~12 years | 835 (18.74%) | 784 (18.89%) | 51 (16.72%) | | |
| | > 12 years | 2165 (48.59%) | 2010 (48.42%) | 155 (50.82%) | | |
| Male age | Mean(SD) | 31.51(4.79) | 31.50(4.78) | 31.60(4.93) | t = −0.36 | 0.716 |
| Male education | < 6 years | 99 (2.22%) | 93 (2.24%) | 6 (1.97%) | χ²=1.47 | 0.690 |
| | 6~9 years | 1412 (31.69%) | 1321 (31.82%) | 91 (29.84%) | | |
| | 10~12 years | 988 (22.17%) | 924 (22.26%) | 64 (20.98%) | | |
| | > 12 years | 1957 (43.92%) | 1813 (43.68%) | 144 (47.21%) | | |
| Infertility years | Median(Q1-Q3) | 3.30 (1.90-5.20) | 3.30 (1.80-5.20) | 3.30 (2.30-5.20) | z = −1.70 | 0.090 |
| Classification of Infertility | Primary | 2405 (53.97%) | 2231 (53.75%) | 174 (57.05%) | χ²=1.25 | 0.264 |
| | Secondary | 2051(46.03%) | 1920 (46.25%) | 131 (42.95%) | | |
| Etiology of Infertility | Tubal factor | 2587 (58.06%) | 2417 (58.23%) | 170 (55.74%) | | 0.122* |
| | Male factor | 25 (0.56%) | 22 (0.53%) | 3 (0.98%) | | |
| | Other factors | 1755 (39.39%) | 1625 (39.15%) | 130 (42.62%) | | |
| | Multi factors | 89 (2.00%) | 87 (2.10%) | 2 (0.66%) | | |
| AMH | Median(Q1-Q3) | 3.99 (2.19-6.53) | 3.99 (2.21-6.51) | 3.87 (1.94-6.88) | z = 0.57 | 0.570# |
| E2 | Median(Q1-Q3) | 40 (27.70-55) | 40 (27.68-55) | 39 (27.78-56) | z = 0.35 | 0.726# |
| FSH | Median(Q1-Q3) | 6.93 (5.74-8.40) | 6.95 (5.77-8.40) | 6.84 (5.50-8.41) | z = 1.58 | 0.114# |
| LH | Median(Q1-Q3) | 4.78 (3.33 - 7.15) | 4.79 (3.35-7.17) | 4.60 (3.17-6.82) | z = 1.44 | 0.150# |
| P | Median(Q1-Q3) | 0.63 (0.39-0.95) | 0.63 (0.39-0.95) | 0.64 (0.41-0.96) | z = −0.87 | 0.387# |
| PRL | Median(Q1-Q3) | 12.22 (8.91-17.08) | 12.17 (8.91-16.86) | 13.61 (9.32-19.14) | z = −3.29 | 0.001# |
| T | Median(Q1-Q3) | 0.52 (0.38 - 0.70) | 0.51 (0.38-0.70) | 0.54 (0.39-0.73) | z = −2.23 | 0.026# |

CNY: Chinese New Year; SD: standard deviation; Q1-Q3: the 1st quartile to the 3rd quartile; AMH, anti-Müllerian hormone; E2, estrogen; FSH, Follicle-Stimulating Hormone; LH, luteinizing hormone; P, progesterone; PRL, prolactin; T, testosterone; *Fisher's exact; #Manne-Whitney Test

two high-quality embryos (23.61%) compared to the N-CNY group (21.39%). Notably, the presence of four high-quality embryos was extremely rare, observed only in the CNY group (0.33%) (Table 2).

No significant variability was noted in the embryonic development stage at the time of transfer (p = 0.284). Most embryos reached day 3 of development, with a higher proportion in the CNY group (80.33%) compared to the N-CNY group (74.95%). Fewer embryos developed to day 5, with 16.67% overall and fewer in the CNY group (13.77%). Embryos reaching day 6 were also more common in the N-CNY group (5.90%) than in the CNY group (4.59%) (Table 2).

The majority of embryo transfers involved two embryos, with 73.20% of all participants undergoing such transfers. This procedure was more prevalent in the CNY group (78.36%) compared to the N-CNY group (72.83%), although the difference was not statistically significant (p = 0.158). Single embryo transfers were less common and notably lower in the CNY group (21.64%) compared to the N-CNY group (27.10%) (p = 0.158) (Table 2).

The location of the embryonic sacs showed no significant variation between the groups (p = 0.541). The majority of sacs were uterine (N-CNY: 52.54%, CNY: 52.79%). Extrauterine and combined intrauterine and extrauterine sacs were rare across all participants. Regarding the number of sacs, there was no significant difference between groups, with the majority having one or two sacs (Table 2).

The rates of biochemical pregnancy were identical across groups at 57.05%. Clinical pregnancy rates were slightly higher in the CNY group (54.43%) compared to the N-CNY group (53.60%), but the difference was not statistically

**Table 2. Treatment protocol and results for all participants.**

| | All (n=4456) | N-CNY (n=4151) | CNY (n=305) | Statistics | P-value |
|---|---|---|---|---|---|
| Treatment | | | | | |
| HRC | 3819 (85.70%) | 3558 (85.71%) | 261 (85.57%) | | 0.927* |
| OSC | 64 (1.44%) | 61 (1.47%) | 3 (0.98%) | | |
| NC | 121 (2.72%) | 113 (2.72%) | 8 (2.62%) | | |
| DR plus HRC | 452 (10.14%) | 419 (10.10%) | 33 (10.82%) | | |
| Protocol | | | | | |
| GnRH antagonist | 1445 (32.43%) | 1330 (32.04%) | 115 (37.70%) | | 0.103* |
| GnRH agonist | 1625 (36.47%) | 1533 (36.93%) | 92 (30.16%) | | |
| Ultra-long GnRH agonist | 1114 (25.00%) | 1037 (24.98%) | 77 (25.25%) | | |
| PPOS | 242 (5.43%) | 224 (5.40%) | 18 (5.90%) | | |
| Others | 30 (0.67%) | 27 (0.65%) | 3 (0.98%) | | |
| Oocyte retrieval season | | | | | |
| Spring (Mar.~May) | 1198 (26.89%) | 1190 (28.67%) | 8 (2.62%) | $x^2$=290.20 | <0.001 |
| Summer (Jun.~Aug.) | 1403 (31.49%) | 1373 (33.08%) | 30 (9.84%) | | |
| Fall (Sept.~Nov.) | 1100 (24.69%) | 931(22.43%) | 169 (55.41%) | | |
| Winter (Dec.~Feb.) | 754 (16.93%) | 657(15.83%) | 98 (32.13%) | | |
| Number of high quality embryos | | | | | |
| 0 | 2152 (48.29%) | 2009 (48.40%) | 143 (46.89%) | | 0.087* |
| 1 | 1343 (30.14%) | 1254 (30.21%) | 89 (29.18%) | | |
| 2 | 960 (21.54%) | 888(21.39%) | 72 (23.61%) | | |
| 4 | 1 (0.02%) | 0 (0.00%) | 1 (0.33%) | | |
| Days | | | | | |
| 2 | 1(0.02%) | 1(0.02%) | 0(0.00%) | | 0.284* |
| 3 | 3356(75.31%) | 3111(74.95%) | 245(80.33%) | | |
| 4 | 97(2.18%) | 93(2.24%) | 4(1.31%) | | |
| 5 | 743(16.67%) | 701(16.89%) | 42(13.77%) | | |
| 6 | 259(5.81%) | 245(5.90%) | 14(4.59%) | | |
| Number of transfer | | | | | |
| 0 | 1(0.02%) | 1(0.02%) | 0(0.00%) | | |
| 1 | 1191 (26.73%) | 1125 (27.10%) | 66 (21.64%) | | 0.158* |
| 2 | 3262 (73.20%) | 3023 (72.83%) | 239 (78.36%) | | |
| 3 | 2 (0.05%) | 2 (0.05%) | 0 (0.00%) | | |
| Sac site | | | | | |
| Unknown | 2066 (46.36%) | 1927 (46.42%) | 139 (45.57%) | | 0.541* |
| Uterine | 2342 (52.56%) | 2181 (52.54%) | 161 (52.79%) | | |
| Extrauterine | 45 (1.01%) | 40 (0.96%) | 5 (1.64%) | | |
| Intrauterine combine extrauterine | 3 (0.07%) | 3 (0.07%) | 0 (0.00%) | | |
| Number of Sac | | | | | |
| 0 | 2110 (47.35%) | 1966 (47.36%) | 144 (47.21%) | | 0.765* |
| 1 | 1611 (36.15%) | 1505 (36.26%) | 106 (34.75%) | | |
| 2 | 731 (16.40%) | 676 (16.29%) | 55 (18.03%) | | |
| 3 | 4 (0.09%) | 4 (0.10%) | 0 (0.00%) | | |
| Biochemical pregnant | 2542 (57.05%) | 2368 (57.05%) | 174 (57.05%) | $x^2$=0.00 | 0.999 |
| Clinical pregnant | 2391 (53.66%) | 2225 (53.60%) | 166 (54.43%) | $x^2$=0.08 | 0.780 |
| Abortion | 393 (8.82%) | 367 (8.84%) | 26 (8.52%) | $x^2$=0.04 | 0.851 |
| Live birth | 1954 (43.85%) | 1819 (43.82%) | 135 (44.26%) | $x^2$=0.02 | 0.881 |

CNY: Chinese New Year; HRC: hormone replace cycle; OSC: Ovulation Stimulating Cycle; NC: Natural Cycle; DR: Down regulation; PPOS: Progesterone-Primed Ovarian Stimulation; *Fisher's exact

significant (p = 0.780). The rates of abortion and live births were also similar between groups, with live births occurring in 43.85% of all cases and slightly higher in the CNY group (44.26%) (Table 2).

## Univariate analysis and multivariate logistic analysis for live birth

In the univariate analysis of fertility treatment outcomes, age was a significant factor for both females and males. Women over the age of 35 had significantly lower odds of treatment success (OR = 0.34, 95% CI: 0.28–0.41, p < 0.001). This trend was also evident in males over 35, who exhibited reduced odds of successful outcomes (OR = 0.42, 95% CI: 0.35–0.50, p < 0.001). Infertility duration also played a critical role, with participants experiencing infertility for 6–10 years showing a slight reduction in success rates (OR = 0.84, 95% CI: 0.73–0.97, p = 0.019). Participants with more than ten years of infertility did not show a statistically significant difference when compared to those with shorter durations (Table 3).

High AMH levels (>3) were associated with a significant increase in the likelihood of positive outcomes (OR = 1.66, 95% CI: 1.46–1.88, p < 0.001). Other hormonal factors, such as progesterone levels greater than one, also indicated higher odds of success (OR = 1.29, 95% CI: 1.12–1.48, p < 0.001). Conversely, higher FSH levels (>10) corresponded to lower success rates (OR = 0.73, 95% CI: 0.61–0.89, p = 0.001) (Table 3).

In the multivariate analysis, adjusting for confounding factors, the impact of female age remained significant, with women over 35 showing decreased odds of success (OR = 0.54, 95% CI: 0.42–0.69, p < 0.001). Male age over 35 also continued to negatively affect treatment outcomes, albeit the association was weaker (OR = 0.75, 95% CI: 0.60–0.94, p = 0.011). The positive impact of high AMH levels was slightly diminished but still significant (OR = 1.21, 95% CI: 1.05–1.41, p = 0.009) (Table 3).

The results also evaluated the effect of various treatment protocols compared to the HRC as a reference. Using OSC was significantly less effective (OR = 0.35, 95% CI: 0.18–0.66, p = 0.001), while the use of GnRH antagonist and follicular phase long-acting GnRH agonist protocols showed improved outcomes compared to HRC.

The analysis demonstrated that the timing of implementation relative to CNY did not significantly impact the likelihood of achieving a live birth. In the univariate analysis, the OR for live birth around CNY was 1.02 (95% CI: 0.81–1.29, p = 0.881), indicating no significant association. Similarly, the multivariate analysis, which adjusted for potential confounders, also showed no significant effect of CNY on live birth outcomes, with an OR of 1.11 (95% CI: 0.86–1.42, p = 0.421) (Table 3).

## Analysis based on the PSM dataset

The baseline analysis compared characteristics between the N-CNY group and the CNY group, each consisting of 292 participants after PSM. There were no significant differences in most baseline characteristics between the two groups. Female ages were comparable, with an average age of 31.15 years in the N-CNY group and 30.75 years in the CNY group (p = 0.308). Educational levels across different categories also showed no significant differences, with the proportions of women with greater than 12 years of education nearly identical in both groups (50.34% in N-CNY vs. 50.68% in CNY). Male participants showed a similar trend, with a mean age of 32.33 years in the N-CNY group and 31.52 years in the CNY group (p = 0.056) (Table 4).

Infertility characteristics such as mean years of infertility and types of infertility (primary vs. secondary) were similarly distributed between the groups. There was also no significant difference in the distribution of infertility etiologies, including tubal factor and male factor (Table 4).

Hormone levels measured included FSH, E2, P, PRL, LH, and AMH, all showing no statistically significant differences between groups. For instance, FSH levels averaged 7.19 μIU/mL in both groups, and AMH levels were slightly higher in the CNY group (4.97 ng/mL) compared to the N-CNY group (4.87 ng/mL), but the difference was not statistically significant (p = 0.981) (Table 4).

**Table 3. Univariate analysis and multivariate logistic analysis for live birth.**

| | | Univariate analysis | | | Multivariate analysis | |
|---|---|---|---|---|---|---|
| | | n (%) | OR (95%CI) | *P*-value | OR (95%CI) | *P*-value |
| Female age | >35 | 649 (14.56%) | 0.34 (0.28, 0.41) | <0.001 | 0.54 (0.42, 0.69) | <0.001 |
| Female education (< 6 years as ref.) | 6~9 years | 1316 (29.53%) | 1.16 (0.81, 1.66) | 0.426 | 1.05 (0.73, 1.53) | 0.777 |
| | 10~12 years | 835 (18.74%) | 1.36 (0.94, 1.97) | 0.098 | 1.25 (0.85, 1.83) | 0.253 |
| | > 12 years | 2165 (48.59%) | 1.35 (0.95, 1.92) | 0.093 | 1.23 (0.85, 1.77) | 0.277 |
| Male age | >35 | 749 (16.81%) | 0.42 (0.35, 0.50) | <0.001 | 0.75 (0.60, 0.94) | 0.011 |
| Male education (< 6 years as ref.) | 6~9 years | 1412 (31.69%) | 0.94 (0.62, 1.42) | 0.773 | | |
| | 10~12 years | 988 (22.17%) | 1.18 (0.78, 1.80) | 0.431 | | |
| | > 12 years | 1957 (43.92%) | 1.20 (0.80, 1.81) | 0.379 | | |
| Infertility years (≤5 years as ref.) | 6~10 years | 1008 (22.62%) | 0.84 (0.73, 0.97) | 0.019 | 0.86 (0.74, 1.00) | 0.053 |
| | >10 year | 187 (4.20%) | 0.55 (0.40, 0.76) | 0.000 | 0.86 (0.61, 1.21) | 0.374 |
| AMH | >3 | 2855 (64.07%) | 1.66 (1.46, 1.88) | <0.001 | 1.21 (1.05, 1.41) | 0.009 |
| E2 | >55 | 1099 (24.66%) | 0.92 (0.80, 1.06) | 0.236 | | |
| FSH | >10 | 508 (11.40%) | 0.73 (0.61, 0.89) | 0.001 | 1.07 (0.86, 1.32) | 0.561 |
| LH | >5 | 2082 (46.72%) | 1.10 (0.98, 1.24) | 0.102 | | |
| P | >1 | 1012 (22.71%) | 1.29 (1.12, 1.48) | 0.000 | 1.25 (1.08, 1.44) | 0.003 |
| PRL | >10 | 2907 (65.24%) | 1.11 (0.98, 1.26) | 0.096 | 0.99 (0.87, 1.12) | 0.828 |
| T | >0.45 | 2707 (60.75%) | 1.10 (0.98, 1.25) | 0.109 | | |
| Classification of Infertility (secondary) | | 2051 (46.03%) | 0.75 (0.66, 0.84) | <0.001 | 0.90 (0.78, 1.02) | 0.100 |
| Etiology of infertility (tubal factor as ref.) | Male factor | 25 (0.56%) | 0.89 (0.40, 1.99) | 0.776 | 0.81 (0.35, 1.85) | 0.619 |
| | Other factors | 1755 (39.39%) | 1.12 (0.99, 1.27) | 0.062 | 1.13 (0.99, 1.28) | 0.072 |
| | Multi factors | 89 (2.00%) | 0.83 (0.53, 1.27) | 0.386 | 1.01 (0.64, 1.59) | 0.979 |
| Treatment regimen (HRC as ref.) | OSC | 64 (1.44%) | 0.28 (0.15, 0.52) | <0.001 | 0.35 (0.18, 0.66) | 0.001 |
| | NC | 121 (2.72%) | 0.88 (0.61, 1.26) | 0.479 | 1.10 (0.75, 1.60) | 0.641 |
| | DR plus HRC | 452 (10.14%) | 0.64 (0.52, 0.78) | <0.001 | 0.82 (0.66, 1.01) | 0.067 |
| Number of high quality embryos | | | 1.03 (0.96, 1.11) | 0.373 | | |
| Number of transfer | | | 1.17 (1.03, 1.34) | 0.020 | 1.13 (0.98, 1.30) | 0.100 |
| Protocol (GnRH antagonist as ref.) | GnRH agonist | 1625 (36.47%) | 1.59 (1.38, 1.84) | <0.001 | 1.38 (1.19, 1.60) | <0.001 |
| | Ultra-long GnRH agonist | 1114 (25.00%) | 1.48 (1.26, 1.74) | <0.001 | 1.19 (1.01, 1.41) | 0.042 |
| | PPOS | 242 (5.43%) | 0.52 (0.38, 0.72) | <0.001 | 0.62 (0.45, 0.86) | 0.005 |
| | Others | 30 (0.67%) | 0.94 (0.44, 1.99) | 0.876 | 1.28 (0.58, 2.82) | 0.539 |
| Oocyte retrieval season (Spring as ref.) | Summer | 1403 (31.49%) | 1.11 (0.95, 1.29) | 0.196 | 1.11 (0.94, 1.30) | 0.217 |
| | Fall | 1100 (24.69%) | 0.97 (0.83, 1.15) | 0.751 | 0.95 (0.80, 1.13) | 0.568 |
| | Winter | 755 (16.94%) | 0.90 (0.75, 1.08) | 0.259 | 0.88 (0.72, 1.06) | 0.176 |
| DAYS (two days as ref.) | 3 | 3356 (75.31%) | 80617.09 (0.00, inf.) | 0.954 | | |
| | 4 | 97 (2.18%) | 44974.08 (0.00, inf.) | 0.957 | | |
| | 5 | 743 (16.67%) | 106311.52 (0.00, inf.) | 0.953 | | |
| | 6 | 259 (5.81%) | 64190.89 (0.00, inf.) | 0.955 | | |
| CNY | | 305 (6.84%) | 1.02 (0.81, 1.29) | 0.881 | 1.11 (0.86, 1.42) | 0.421 |

CNY: Chinese New Year; OR: odds ratio; CI: confidential interval; HRC: hormone replace cycle; OSC: Ovulation Stimulating Cycle; NC: Natural Cycle; DR: Down regulation; PPOS: Progesterone-Primed Ovarian Stimulation; AMH, anti-Müllerian hormone; E2, estrogen; FSH, Follicle-Stimulating Hormone; LH, luteinizing hormone; P, progesterone; PRL, prolactin; T, testosterone

**Table 4. Baseline characters analysis based on the propensity score matching (PSM) dataset.**

| | | N-CNY (n = 292) | CNY (n = 292) | P-value | SMD |
|---|---|---|---|---|---|
| Female age | Mean (SD) | 31.15 (4.71) | 30.75 (4.70) | 0.308 | 0.084 |
| Female education (%) | < 6 years | 6 (2.05) | 12 (4.11) | 0.474 | 0.131 |
| | 6~9 years | 92 (31.51) | 83 (28.42) | | |
| | 10~12 years | 47 (16.10) | 49 (16.78) | | |
| | > 12 years | 147 (50.34) | 148 (50.68) | | |
| Male age | Mean (SD) | 32.33 (5.44) | 31.52 (4.84) | 0.056 | 0.158 |
| Male education (%) | < 6 years | 5 (1.71) | 6 (2.05) | 0.688 | 0.101 |
| | 6~9 years | 93 (31.85) | 88 (30.14) | | |
| | 10~12 years | 51 (17.47) | 62 (21.23) | | |
| | > 12 years | 143 (48.97) | 136 (46.58) | | |
| Infertility years | Mean (SD) | 3.96 (3.05) | 3.85 (2.48) | 0.539 | 0.037 |
| FSH | Mean (SD) | 7.19 (2.74) | 7.19 (3.50) | 0.692 | 0.003 |
| E2 | Mean (SD) | 51.59 (54.62) | 47.97 (60.69) | 0.134 | 0.063 |
| P | Mean (SD) | 0.85 (1.05) | 0.81 (0.80) | 0.565 | 0.045 |
| PRL | Mean (SD) | 14.40 (7.14) | 14.87 (6.96) | 0.347 | 0.067 |
| LH | Mean (SD) | 5.82 (4.19) | 5.70 (4.12) | 0.73 | 0.03 |
| T | Mean (SD) | 0.58 (0.29) | 0.57 (0.26) | 0.777 | 0.029 |
| AMH | Mean (SD) | 4.87 (3.88) | 4.97 (4.14) | 0.981 | 0.023 |
| Classification of Infertility | primary | 155 (53.08) | 165 (56.51) | 0.454 | 0.069 |
| | secondary | 137 (46.92) | 127 (43.49) | | |
| Etiology of Infertility | Tubal factor | 169 (57.88) | 166 (56.85) | 0.265 | 0.165 |
| | Male factor | 8 (2.74) | 2 (0.68) | | |
| | Other factors | 113 (38.70) | 122 (41.78) | | |
| | Multi factors | 2 (0.68) | 2 (0.68) | | |
| Treatment (%) | HRC | 250 (85.62) | 252 (86.30) | 0.271 | 0.164 |
| | OSC | 0 (0.00) | 3 (1.03) | | |
| | NC | 11 (3.77) | 7 (2.40) | | |
| | DR plus HRC | 31 (10.62) | 30 (10.27) | | |
| High quality embryo (%) | 0 | 144 (49.32) | 137 (46.92) | 0.799 | 0.055 |
| | 1 | 84 (28.77) | 85 (29.11) | | |
| | 2 | 64 (21.92) | 70 (23.97) | | |
| Number of transfer (%) | 1 | 67 (22.95) | 66 (22.60) | 1 | 0.008 |
| | 2 | 225 (77.05) | 226 (77.40) | | |
| Protocol (%) | GnRH antagonist | 104 (35.62) | 108 (36.99) | 0.888 | 0.088 |
| | GnRH agonist | 87 (29.79) | 91 (31.16) | | |
| | Ultra-long GnRH agonist | 76 (26.03) | 74 (25.34) | | |
| | PPOS | 22 (7.53) | 16 (5.48) | | |
| | Others | 3 (1.03) | 3 (1.03) | | |
| Retrieval season (%) | Spring | 6 (2.05) | 7 (2.40) | 0.775 | 0.087 |
| | Summer | 32 (10.96) | 30 (10.27) | | |
| | Fall | 168 (57.53) | 158 (54.11) | | |
| | Winter | 86 (29.45) | 97 (33.22) | | |
| DAYS (%) | 3 | 229 (78.42) | 233 (79.79) | 0.899 | 0.063 |
| | 4 | 5 (1.71) | 4 (1.37) | | |
| | 5 | 46 (15.75) | 41 (14.04) | | |
| | 6 | 12 (4.11) | 14 (4.79) | | |

CNY: Chinese New Year; SD: standard deviation; SMD: standard mean difference; HRC: hormone replace cycle; OSC: Ovulation Stimulating Cycle; NC: Natural Cycle; DR: Down regulation; PPOS: Progesterone-Primed Ovarian Stimulation; AMH, anti-Müllerian hormone; E2, estrogen; FSH, Follicle-Stimulating Hormone; LH, luteinizing hormone; P, progesterone; PRL, prolactin; T, testosterone

Treatment modalities assessed included HRC, OSC, and NC, among others. Again, the distribution of these treatments showed no significant variance between the two groups (Table 4).

The analysis of live birth rates between the N-CNY group and the CNY group indicated that the proportion of live births was higher in the CNY group (44.18%) compared to the N-CNY group (39.04%), with the respective proportions reflecting a non-significant chi-square value (p = 0.208) (Table 5).

## Discussion

From the results, conducting frozen embryo transfers during the Spring Festival does not affect live birth outcomes. This finding is inconsistent with previous research, which suggested that fresh embryo transfers during the Spring Festival could impact pregnancy outcomes [2]. The reasons for these results are analyzed as follows:

The absence of a significant difference in live birth outcomes between frozen embryo transfers during the Spring Festival and other times of the year adds an intriguing dimension to our understanding of ART outcomes and the influence of sociocultural factors [16,17]. While previous studies have highlighted the potential negative impact of the Chinese New Year on fresh embryo transfers, our findings suggest that frozen embryo transfers might be insulated from these effects to some extent. This resilience could stem from the nature of frozen cycles themselves, which are less involved and shorter in duration than fresh cycles, potentially reducing the window for stress and sociocultural activities to impact the outcome [18].

Moreover, the role of psychological stress in ART outcomes remains a complex and somewhat underexplored area [19–21]. Our study suggests that the processes involved in frozen embryo transfers might not be as vulnerable to psychological stress as those in fresh cycles, possibly due to the shorter duration and less intensive preparation required. This hypothesis is consistent with the theory that certain stages of the IVF process may be more sensitive to stress than others [22], an area that warrants further investigation.

The frozen embryo transfer cycle is relatively shorter than that of the fresh transfer cycle [23], mainly involving the process of embryo implantation, which may be less affected by psychological factors [24,25]. Compared to the process of ovarian stimulation, which might be more susceptible to the influence of psychological factors [26,27]. This also indicates that the process of preparing the endometrium is not as sensitive to the influence of psychological factors as the process of embryo formation [28].

Our findings reveal no significant differences in oocyte retrieval among four different seasons. In our study, oocyte retrievals were predominantly conducted during autumn and winter in the CNY group, whereas they occurred consistently across all four seasons in the N-CNY group. This disparity contrasts with the conclusions drawn from Correia's research [29]. Correia's study suggests that the average daily temperature on the day of oocyte retrieval, rather than the embryo transfer, correlates with the live birth rate. These differences may be attributed to variations in the data included in their multivariate logistic analysis. While their models account for age, quadratic age, and average temperature on the day of oocyte retrieval and on the day of frozen embryo transfer, it's important to note that factors influencing live birth rates extend beyond these variables alone. Factors such as endometrial thickness, ovarian stimulation protocols, embryo quality, and infertility duration should also be included in the analysis. Thus, when accounting for these additional factors comprehensively, our study did not observe a significant impact of the egg retrieval season on the live birth rate. Naturally, further discussions on this matter can be pursued with more extensive data in the future.

**Table 5. The live birth rate between the two groups.**

|  | n | Proportion | 95%CI | $\chi^2$ | *P*-value |
|---|---|---|---|---|---|
| N-CNY | 114 | 0.3904 | 0.3359 - 0.4478 | 1.59 | 0.208 |
| CNY | 129 | 0.4418 | 0.3856 - 0.4995 |  |  |

CNY: Chinese New Year.

Our research indicates that both female and male ages independently influence the live birth rate when the age exceeds 35. Additionally, AMH level and infertility duration are an independent influencing factor on live birth rates. These findings align with previous research that identified predictors of IVF success [30].

According to our research, the optimal protocol for FET is HRT, which contradicts existing research [31]. Lou L suggested that letrozole-stimulated cycles offer a significant advantage over HRT and natural cycles. This disparity may stem from differences in the patient populations studied. Lou L included women with normal menstrual cycles, whereas our study encompassed various patient profiles under the Patient-Oriented Strategies Encompassing Individualized Oocyte Number (POSEDON) criteria. Moreover, Lou L's research did not consider stimulation protocols before oocyte retrieval. In contrast, our study incorporated four stimulation protocols into multivariate logistic analysis, revealing that luteal phase GnRH agonist and follicular phase long-acting GnRH agonist protocols were superior to the GnRH antagonist protocol in terms of live birth rate, while the PPOS protocol was inferior to the GnRH antagonist protocol. This finding corroborates our previous research, indicating that the follicular phase long-acting GnRH agonist protocol yields better embryo quality than the GnRH antagonist protocol [32].

In the DR plus HRT protocol, data indicates that the best scenario shows a 1% increase in live births compared to HRC, while the worst scenario shows a 34% decrease, with an effect size of an 18% reduction in live births. Statistically, an increase in sample size would result in a significant statistical difference. We selected patients in their first transfer cycles, and these were FET cycles. Clinically, the DR plus HRT protocol is primarily used in the first FET cycle for patients with endometriosis or following treatment for endometritis or excessive endometrial proliferation. Therefore, there is a selection bias in the population, which results in no significant statistical difference.

## Limitations

It is important to consider the limitations of our study, including its retrospective design and the focus on a single center's population. These factors may limit the generalizability of our findings. The small sample size of patients undergoing frozen embryo transfer during the Spring Festival could also diminish the statistical power of our analysis, potentially obscuring subtle effects that a larger study might uncover.

Future research should aim to replicate these findings in larger, multicenter studies to explore the generalizability of our results. Additionally, prospective studies designed to measure psychological stress levels directly could provide valuable insights into how stress affects different stages of the IVF process. Investigating the impact of other significant sociocultural events and holidays within different cultural contexts could also shed light on the universality of our findings. Such studies would contribute significantly to our understanding of the intricate balance between medical procedures and the psychological and sociocultural environment in which they are carried out.

## Conclusions

Finally, our findings underscore the need for comprehensive patient support during ART procedures. While our study suggests that frozen embryo transfers may be less susceptible to the effects of sociocultural events like the Spring Festival, it is crucial to acknowledge and address the psychological needs of all patients undergoing IVF treatments. Tailoring support services to the specific phases of the IVF cycle and recognizing the potential stressors unique to each patient's cultural and social context could enhance ART outcomes and patient wellbeing.

## Author contributions

**Conceptualization:** Jiajia Zhai, Guimin Hao.

**Data curation:** Jiajia Zhai, Aimin Li, Shibin Zhao.

**Formal analysis:** Junyan Zhang.

**Investigation:** Jiajia Zhai, Shibin Zhao.

**Methodology:** Junyan Zhang.

**Supervision:** Guimin Hao.

**Validation:** Bozheng Zhang.

**Writing – original draft:** Songping Yi.

**Writing – review & editing:** Jiajia Zhai, Songping Yi, Junyan Zhang, Aimin Li, Shibin Zhao, Bozheng Zhang, Guimin Hao.

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
