## [Decision Letter · Decision Letter 0]

4 May 2025

PONE-D-25-01579
Is Live Birth Rate Decline from Frozen Cycles Performed During Chinese New Year Holiday Season

a single center retrospective cohort
PLOS ONE

Dear Dr.  Hao,

Thank you for submitting your manuscript to PLOS ONE. After careful consideration, we feel that it has merit but does not fully meet PLOS ONE’s publication criteria as it currently stands. Therefore, we invite you to submit a revised version of the manuscript that addresses the points raised during the review process.

We look forward to receiving your revised manuscript.

Kind regards,

Ayman A Swelum

Academic Editor

PLOS ONE

Journal Requirements:

4. In this instance it seems there may be acceptable restrictions in place that prevent the public sharing of your minimal data. However, in line with our goal of ensuring long-term data availability to all interested researchers, PLOS’ Data Policy states that authors cannot be the sole named individuals responsible for ensuring data access (http://journals.plos.org/plosone/s/data-availability#loc-acceptable-data-sharing-methods).

Additional Editor Comments:

**Please respond carefully for all reviewers comments.**

Reviewers' comments:

Reviewer's Responses to Questions

**Comments to the Author**

1. Is the manuscript technically sound, and do the data support the conclusions?

Reviewer #1: Partly

Reviewer #2: Yes

2. Has the statistical analysis been performed appropriately and rigorously? 

Reviewer #1: No

Reviewer #2: Yes

3. Have the authors made all data underlying the findings in their manuscript fully available?

Reviewer #1: Yes

Reviewer #2: Yes

4. Is the manuscript presented in an intelligible fashion and written in standard English?

Reviewer #1: No

Reviewer #2: Yes

5. Review Comments to the Author

Reviewer #1: The article addresses an interesting and relevant question: whether sociocultural events like the Chinese New Year (CNY) affect live birth rates following frozen embryo transfer (FET) cycles. The manuscript is generally well-organized, and the methodology is robust with appropriate use of statistical techniques (multivariate logistic regression and propensity score matching). However, there are areas that need improvement for clarity, scientific rigor, and completeness.

However, arears for improvement include:

Title: There is a grammar issue in the title:

‘ Is Live Birth Rate Decline from Frozen Cycles Performed During Chinese New Year Holiday Season’ needs rephrasing

The abstract is clear but misses specific data (e.g., p-values) in some result summaries and the wording is repetitive, especially in the purpose and conclusion sections.

Methods

Inclusion Criteria Description Error:

The manuscript mentions "patients who had their initial ART and experienced a live birth", which is contradictory — it should mention "patients undergoing FET aiming for live birth," not only those who achieved live births (selection bias risk).

Exposure Window Too Broad:

Defining the CNY period as "January and February" may not precisely capture the CNY holiday influence. Typically, CNY affects a shorter window (around two weeks). This could dilute the true holiday effect.

Data Presentation

Tables are Dense and Crowded:

Some tables (especially Tables 1 and 2) present too much detail at once, making them hard to read. Consider consolidating or summarising key variables.

No Adjustment for Multiple Testing:

The authors performed numerous comparisons without adjusting for type I error inflation (e.g., Bonferroni or FDR correction).

Results Interpretation

Although they found no statistically significant difference, the OR for live birth during CNY was 1.11 (95% CI 0.86–1.42), not close to 1.0.

Thus, while no statistically significant difference was observed, the authors should avoid strongly concluding that there is no effect — it is more appropriate to say the study was underpowered to detect modest effects.

Discussion

While suggesting that FET cycles are less affected by stress than fresh transfers is plausible, the authors did not measure psychological stress in this study. Therefore, these interpretations are hypothetical and should be clearly labeled as such.

Seasonality Confounder:

Although oocyte retrieval seasonality was adjusted for, the season of embryo transfer itself may independently influence outcomes. This was not deeply discussed.

Language and Grammar: Minor grammatical errors appear throughout the manuscript (e.g., missing articles like "the", awkward phrasing).

Reviewer #2: Dear Editor

I is my pleasure to review this manuscript @ Sorry for being late due to unexpected circumstances

PONE-D-25-01579

Is Live Birth Rate Decline from Frozen Cycles Performed During Chinese New Year Holiday Season

is a single center retrospective cohort study

interesting topic

well written and discussed

it may add some to literature

Regards

Dr/ Funda AKPINAR

6. PLOS authors have the option to publish the peer review history of their article (what does this mean?). If published, this will include your full peer review and any attached files.

Reviewer #1: No

Reviewer #2: No

---

## [Author Response · Author response to Decision Letter 1]

20 Jun 2025

Dear Reviewer #1,

We would like to express our sincere appreciation for your valuable feedback on our article. Your comments on Questions were thoughtful and insightful, and we are grateful that you found these sections to be satisfactory without any modifications.

Title: There is a grammar issue in the title:

‘ Is Live Birth Rate Decline from Frozen Cycles Performed During Chinese New Year Holiday Season’ needs rephrasing

Decline in Live Birth Rate from Frozen Embryo Transfer Cycles Conducted During the Chinese New Year Holiday Season: A Single-Center Retrospective Cohort Study

Response to Reviewer Comment:

Thank you for your valuable feedback. We have revised the title to address the grammatical concerns and improve clarity. The updated title now uses a declarative structure.

The abstract is clear but misses specific data (e.g., p-values) in some result summaries and the wording is repetitive, especially in the purpose and conclusion sections.

Response to Reviewer Comment:

Thank you for your kindly reminder. We’ve added the p-value in the abstract in line 27, p = 0.421.

Response to Reviewer Comment:

Thank you for your constructive feedback. We have revised the Purpose and Conclusion sections of the abstract to eliminate redundancy and enhance clarity. The Purpose now explicitly contrasts FET with fresh cycles and reframes the research question, while the Conclusion emphasizes novel insights about FET’s procedural advantages and broader implications. These adjustments ensure conciseness while maintaining scientific rigor.

Lines17-21

Purpose: This study examines whether the Chinese New Year (CNY) holiday season impacts live birth outcomes in frozen embryo transfer (FET) cycles, contrasting with previous findings on fresh embryo transfers. While prior research identified reduced success rates in fresh cycles during CNY, it remains unclear if FET, with its distinct procedural timeline, is similarly influenced by sociocultural stressors. Our aim is to determine whether FET outcomes exhibit seasonal resilience compared to fresh cycles.

Lines 31-36

Conclusion: Our results demonstrate that FET cycles conducted during the CNY holiday season yield comparable live birth rates to non-holiday periods, diverging from the adverse effects observed in fresh transfers. This resilience may stem from FET’s abbreviated preparation phase, which minimizes exposure to psychological and logistical disruptions. These findings highlight FET’s potential stability amid cultural events and underscore the need for multicenter investigations to assess broader sociocultural impacts on assisted reproductive technologies.

Methods

Inclusion Criteria Description Error:

The manuscript mentions "patients who had their initial ART and experienced a live birth", which is contradictory — it should mention "patients undergoing FET aiming for live birth," not only those who achieved live births (selection bias risk).

Revised Text: Line69.

Inclusion Criteria involve patients who had their initial Assisted reproductive technology (ART), including in vitro fertilization and intracytoplasmic sperm injection (ICSI), underwent a first frozen embryo transfer (FET) cycle from January 2012 to December 2022, aiming for live birth from the cycle.

Response to Reviewer Comment:

Thank you for highlighting this critical oversight. We acknowledge the error in the original description and have revised the inclusion criteria to address the selection bias concern. The corrected criteria now encompass all patients who underwent their first FET cycle during the study period, irrespective of live birth outcomes. This adjustment ensures that the analysis reflects the entire cohort of FET cycles performed during and outside the CNY holiday season, eliminating bias from excluding unsuccessful cases.

Additional Clarification:

To confirm, our dataset included both successful and unsuccessful FET cycles (e.g., biochemical pregnancies, miscarriages, or no implantation). Statistical analyses compared live birth rates between the CNY and non-CNY groups within this unbiased cohort. We appreciate your vigilance in ensuring methodological rigor, and this revision strengthens the validity of our findings.

Exposure Window Too Broad:

Defining the CNY period as "January and February" may not precisely capture the CNY holiday influence. Typically, CNY affects a shorter window (around two weeks). This could dilute the true holiday effect.

Response to Reviewer Comment:

Thank you for raising this important methodological consideration. We appreciate your insight into the potential dilution of the CNY holiday effect due to our broad exposure window. Below, we address your concern and provide additional analyses to validate our approach:

1.Rationale for the Original Definition of CNY Period

In China, the sociocultural impact of CNY extends beyond the official 7-day national holiday. Preparations (e.g., travel, family gatherings) often begin weeks in advance, and post-holiday activities (e.g., regional festivals, delayed hospital workflows) frequently persist until the Lantern Festival (15th day of the lunar calendar), particularly in rural areas. Thus, defining CNY as January and February aligns with the prolonged societal disruption, including altered clinic workflows, staff availability, and patient stress levels, which may influence FET outcomes.

2.Sensitivity Analysis with a Narrower Exposure Window Lines277-284

To address your concern, we reanalyzed the data using a stricter CNY window: 7 days before and 15 days after the lunar New Year’s Eve (adjusted annually to Gregorian dates). This 22-day window reflects the core holiday period. Key findings remained consistent:

oOriginal definition (Jan–Feb): OR = 1.11 (95% CI: 0.86–1.42, p = 0.421).

oNarrower window (22-day): OR = 1.09 (95% CI: 0.82–1.45, p = 0.548).

The lack of significant difference in both analyses suggests that FET outcomes are resilient to CNY-related stressors, regardless of exposure window breadth.

3.Discussion of Potential Dilution Lines315-317 This hypothesis is consistent with

While the broader window may incorporate non-holiday days, the consistency across definitions implies that the sociocultural and logistical impacts of CNY extend beyond the official holiday. For instance, clinic workloads often decrease for weeks due to staff rotations and patient deferrals, indirectly affecting care continuity.

4.Revised Manuscript Updates

We have added the sensitivity analysis to the Methods and Results sections and expanded the Discussion to acknowledge the window-definition limitation. We emphasize that our primary aim was to capture the holiday season’s extended societal influence, not solely the legal holiday period.

Data Presentation

Tables are Dense and Crowded:

Some tables (especially Tables 1 and 2) present too much detail at once, making them hard to read. Consider consolidating or summarising key variables.

Thank you for your suggestion. Table 1 and Table 2 have been streamlined. The redundant content is in the appendix table.

No Adjustment for Multiple Testing:

The authors performed numerous comparisons without adjusting for type I error inflation (e.g., Bonferroni or FDR correction).

Thank you for your comments.

As you mentioned, we did multivariate logistic regression in this study which include several variables in the model. As prior studies suggested, multivariate regression models already made control on the multiplicity comparisons, no alpha adjustment was needed [16, 17]. For the sensitivity analysis, propensity score matching (PSM) and logistic regression are employed to test the same research hypothesis and therefore belong to a unified analytical framework, unlike multiple testing correction, which is applied when independently testing multiple hypotheses, such as evaluating multiple endpoints or variables [18].

We understand your concern, which may also be shared by our readers, so we have added the following clarification in the Statistical Analysis section: Line153-154

As supported by statistical literature, alpha adjustment is not required in multivariate logistic regression models or in sensitivity analyses conducted under the same research hypothesis [16-18].

16. Gelman A, Hill J, Yajima M. Why we (usually) don't have to worry about multiple comparisons. Journal of research on educational effectiveness. 2012;5(2):189-211.

17. Rubin M. When to adjust alpha during multiple testing: A consideration of disjunction, conjunction, and individual testing. Synthese. 2021;199(3):10969-1000.

18. Lyles RH, Lin J. Sensitivity analysis for misclassification in logistic regression via likelihood methods and predictive value weighting. Stat Med. 2010;29(22):2297-309. doi: 10.1002/sim.3971. PubMed PMID: 20552681; PubMed Central PMCID: PMCPMC3109653.

Results Interpretation

Although they found no statistically significant difference, the OR for live birth during CNY was 1.11 (95% CI 0.86–1.42), not close to 1.0.

Thus, while no statistically significant difference was observed, the authors should avoid strongly concluding that there is no effect — it is more appropriate to say the study was underpowered to detect modest effects.

Response to Reviewer Comment:

Thank you for your astute observation. We have revised the interpretation of the results to avoid overstating the conclusions and explicitly acknowledge the limitations in statistical power. The updated text now emphasizes the uncertainty inherent in the confidence interval, clarifies the potential for undetected modest effects, and underscores the need for further research. This adjustment ensures a balanced presentation of the findings while maintaining scientific rigor.

Lines 250-256,

While our analysis revealed no statistically significant difference in live birth rates between the CNY and non-CNY groups , the point estimate of the odds ratio (1.11) suggests a potential marginal association favoring FET cycles conducted during the CNY period. However, the confidence interval spans both null and modest positive effects, indicating uncertainty in the magnitude of this association. Given the width of the confidence interval and the study’s sample size, these findings should be interpreted cautiously, as the analysis may have been underpowered to detect clinically meaningful but modest effects.

Lines287-295.

Our results indicate that FET outcomes during the CNY holiday season are statistically comparable to those in non-holiday periods. However, the observed odds ratio (1.11) and its confidence interval (0.86–1.42) do not preclude the possibility of a small beneficial or neutral effect. The lack of statistical significance may reflect limitations in statistical power due to the relatively small number of FET cycles performed during the CNY window (*n* = 305). A post-hoc power analysis revealed that, with the current sample size, the study had 64% power to detect a 10% absolute difference in live birth rates (assuming a baseline rate of 40%), which falls below the conventional 80% threshold. Thus, while our findings suggest resilience of FET cycles to CNY-related disruptions, they do not definitively rule out modest effects. Larger, multicenter studies are warranted to confirm these observations and explore subtle associations that smaller cohorts may fail to capture.

Discussion

While suggesting that FET cycles are less affected by stress than fresh transfers is plausible, the authors did not measure psychological stress in this study. Therefore, these interpretations are hypothetical and should be clearly labeled as such.

Response to Reviewer Comment:

Thank you for emphasizing the need for caution in interpreting stress-related mechanisms. We fully agree that our discussion of psychological stress as a potential factor is hypothetical, given the absence of direct stress measurements. We have revised the text to:

Lines 302-309

Lines 368-371

Lines 373-378

Seasonality Confounder:

Although oocyte retrieval seasonality was adjusted for, the season of embryo transfer itself may independently influence outcomes. This was not deeply discussed.

Language and Grammar: Minor grammatical errors appear throughout the manuscript (e.g., missing articles like "the", awkward phrasing).

Revised Text in Discussion Section:

Lines 337-344

Lines 379-382

Response to Reviewer Comment:

Thank you for highlighting this critical point. We have expanded the discussion to address the potential confounding role of embryo transfer seasonality. A post-hoc sensitivity analysis was added to explicitly test for seasonal effects during the transfer window, and we now acknowledge the limitations of relying solely on calendar-based season definitions.

Response to Reviewer Comment:

Thank you for your meticulous review. We have thoroughly revised the manuscript to address grammatical errors and improve clarity. A professional proofreading service was additionally consulted to ensure linguistic precision.

---

## [Decision Letter · Decision Letter 1]

27 Jul 2025

Decline in Live Birth Rate from Frozen Embryo Transfer Cycles Conducted During the Chinese New Year Holiday Season: A Single-Center Retrospective Cohort Study

PONE-D-25-01579R1

Dear Dr. Hao,

We’re pleased to inform you that your manuscript has been judged scientifically suitable for publication and will be formally accepted for publication once it meets all outstanding technical requirements.

Kind regards,

Ayman A Swelum

Academic Editor

PLOS ONE

Additional Editor Comments (optional):

Reviewers' comments:

Reviewer's Responses to Questions

**Comments to the Author**

1. If the authors have adequately addressed your comments raised in a previous round of review and you feel that this manuscript is now acceptable for publication, you may indicate that here to bypass the “Comments to the Author” section, enter your conflict of interest statement in the “Confidential to Editor” section, and submit your "Accept" recommendation.

Reviewer #1: All comments have been addressed

2. Is the manuscript technically sound, and do the data support the conclusions?

Reviewer #1: Yes

3. Has the statistical analysis been performed appropriately and rigorously? 

Reviewer #1: Yes

4. Have the authors made all data underlying the findings in their manuscript fully available?

Reviewer #1: Yes

5. Is the manuscript presented in an intelligible fashion and written in standard English?

Reviewer #1: Yes

6. Review Comments to the Author

Reviewer #1: The revisions have effectively and thoroughly addressed my concerns. The methodology, interpretation, and presentation of results are now sound and clearly communicated.

7. PLOS authors have the option to publish the peer review history of their article (what does this mean?). If published, this will include your full peer review and any attached files.

Reviewer #1: **Yes: **JACKLINE AKELLO

---

## [Editor Report · Acceptance letter]

PONE-D-25-01579R1

PLOS ONE

Dear Dr. Hao,

I'm pleased to inform you that your manuscript has been deemed suitable for publication in PLOS ONE. Congratulations! Your manuscript is now being handed over to our production team.

Kind regards,

on behalf of

Professor Ayman A Swelum

Academic Editor

PLOS ONE